# Molecular sieving of iso-butene from $C_4$ olefins with simultaneous high 1,3-butadiene and n-butene uptakes

Junhui Liu[1], Hanting Xiong[1], Hua Shuai[1], Xing Liu[1], Yong Peng[1], Lingmin Wang[1], Pengxiang Wang[1], Zhiwei Zhao[1], Zhenning Deng[1], Zhenyu Zhou[1], Jingwen Chen [1], Shixia Chen[1], Zheling Zeng[1], Shuguang Deng [2] & Jun Wang [1] ✉

Iso-butene (iso-$C_4H_8$) is an important raw material in chemical industry, whereas its efficient separation remains challenging due to similar molecular properties of $C_4$ olefins. The ideal adsorbent should possess simultaneous high uptakes for 1,3-butadiene ($C_4H_6$) and n-butene (n-$C_4H_8$) counterparts, endowing high efficiency for iso-$C_4H_8$ separation in adsorption columns. Herein, a sulfate-pillared adsorbent, SOFOUR-DPDS-Ni (DPDS = 4,4′-dipyridyldisulfide), is reported for the efficient iso-$C_4H_8$ separation from binary and ternary $C_4$ olefin mixtures. The rigidity in pore sizes and shapes of SOFOUR-DPDS-Ni exerts the molecular sieving of iso-$C_4H_8$, while exhibiting high $C_4H_6$ and n-$C_4H_8$ uptakes. The benchmark Henry's selectivity for $C_4H_6$/iso-$C_4H_8$ (2321.8) and n-$C_4H_8$/iso-$C_4H_8$ (233.5) outperforms most reported adsorbents. Computational simulations reveal the strong interactions for $C_4H_6$ and n-$C_4H_8$. Furthermore, dynamic breakthrough experiments demonstrate the direct production of high-purity iso-$C_4H_8$ (>99.9%) from $C_4H_6$/iso-$C_4H_8$ (50/50, $v/v$), n-$C_4H_8$/iso-$C_4H_8$ (50/50, $v/v$), and $C_4H_6$/n-$C_4H_8$/iso-$C_4H_8$ (50/15/35, $v/v/v$) gas-mixtures.

Iso-butene (iso-$C_4H_8$) is a crucial feedstock for the production of butyl rubber, tert-butanol, and methyl tert-butyl ether (MTBE), with an annual consumption exceeding 30 million tons[1]. Generally, the steam cracking of naphtha generates $C_4$ hydrocarbon mixtures containing 30-60% 1,3-butadiene ($C_4H_6$), 10-20% n-butene (n-$C_4H_8$), and 10-30% iso-$C_4H_8$[2]. While, the stringent purity requirement for polymer-grade iso-$C_4H_8$ (>99.5%) necessitates mandatory purification processes[3]. In industry, $C_4H_6$ is removed from $C_4$ hydrocarbons in high extractive distillation towers (more than 110 trays) at harsh conditions of 3 bar and 323-393 K[4]. Furthermore, thermal-derived methods are inadequate for separating n-$C_4H_8$ and iso-$C_4H_8$ due to their subtle difference in boiling points (0.6 °C, Supplementary Table 2)[3]. The high-purity iso-$C_4H_8$ is obtained through the cracking of MTBE, the reaction product of iso-$C_4H_8$ fraction in $C_4$ mixtures and methanol using strong acidic

ion exchange resins as catalysts[2]. The excessive consumption of organic solvents and energy in traditional separation methods emphasizes the need for an energy- and cost-efficient strategy for iso-$C_4H_8$ purification from $C_4H_6$ and n-$C_4H_8$.

Physisorption utilizing porous adsorbents, e.g., zeolites and metal-organic frameworks (MOFs)[5–13], shows great promise in various challenging gas separations, including $C_2H_2$/$C_2H_4$, $CO_2$/$C_2H_2$, and n-$C_4H_{10}$/iso-$C_4H_{10}$[14–22]. The development of advanced adsorbents capable of recognizing the subtle differences in molecular shapes, sizes, and properties among $C_4$ counterparts remain a formidable challenge (Supplementary Fig. 1 and Supplementary Table 2). Namely, zeolite DD3R exhibited unsatisfactory adsorption capacities (0.832 mmol g$^{-1}$ for $C_4H_6$), resulting in low separation efficiency[15]. In recent examples, adsorbents indiscriminately adsorbed $C_4$

[1]Chemistry and Chemical Engineering School, Nanchang University, Nanchang 330031 Jiangxi, China. [2]School for Engineering of Matter, Transport and Energy, Arizona State University, Tempe, AZ 85287, USA. ✉e-mail: jwang7@ncu.edu.cn

components with unsaturated bonds through open metal sites and high-polar pillars, causing significant co-adsorption and low iso-$C_4H_8$ recovery in fixed-bed columns[23,24]. The ideal adsorbent for iso-$C_4H_8$ purification from $C_4$ olefins should possess high simultaneous adsorption capacities for both $C_4H_6$ and n-$C_4H_8$ while preventing the co-adsorption of iso-$C_4H_8$ in order to achieve optimal separation efficiency (Fig. 1a). To date, achieving complete molecular sieving of iso-$C_4H_8$ remains challenging for MOF adsorbents[1,24]. For example, Prof. Eddaoudi's group reported Y-fum with a suitable aperture size of ~4.7 Å exhibited molecular sieving capabilities for iso-$C_4H_{10}$ and iso-$C_5H_{12}$ from their corresponding n-paraffins[25]. The successful synthesis of this type of adsorbent remains a formidable challenge, which is rarely reported thus far[26-28].

Sulfate anions ($SO_4^{2-}$) possessing abundant lone pair electrons facilitate the formation of coordination bonds with metal ions[29]. Zaworotko et al., reported the first sulfate-pillared hybrid ultramicroporous adsorbent (SOFOUR-1-Zn) in 2021[30]. Our group has recently demonstrated the exceptional molecule recognition ability of $SO_4^{2-}$ groups in a sulfate-pillared adsorbent (SOFOUR-TEPE-Zn, TEPE = 1,1,2,2-tetra(pyridin-4-yl) ethene), which exhibited a benchmark selectivity of 16,833 for $C_2H_2/CO_2$ separation[31]. For separation purposes, the rotation of linear pillars (e.g., $SiF_6^{2-}$, $TiF_6^{2-}$, $GeF_6^{2-}$) in anion-pillared adsorbents induced by interactions with adsorbates may compromise the yield and purity of iso-$C_4H_8$ due to its potential entrance and co-adsorption[20,24]. Moreover, the densely packed pore channels resulting from layer-to-layer sliding after solvent removal may lead to high gate-opening pressures, causing poor adsorption capacity and limited diffusion rate (Fig. 1b)[32,33]. In sharp contrast, the inflexibility of tetrahedral $SO_4^{2-}$ anions enables the preservation of pore shapes and sizes during activation and adsorption processes[34], thereby conferring limited iso-$C_4H_8$ adsorption/diffusion in adsorbents.

Herein, we report a $SO_4^{2-}$-anion pillared adsorbent, SOFOUR-DPDS-Ni (DPDS = 4,4'-dipyridyldisulfide), for efficient iso-$C_4H_8$ separation from $C_4$ olefins. The adsorption isotherms demonstrate simultaneous high uptakes of $C_4H_6$ and n-$C_4H_8$ while almost excluding iso-$C_4H_8$. Notably, the $C_4H_6$ adsorption capacity reaches a high value of 1.15 mmol g$^{-1}$ at 0.001 bar (1000 ppm), indicating its potential for removing trace amounts of $C_4H_6$ from $C_4$ olefin mixtures. The Henry's selectivity is exceptional with values of 2321.8 and 233.5 for $C_4H_6$/iso-$C_4H_8$ and n-$C_4H_8$/iso-$C_4H_8$ at 298 K. Computational simulations confirm that $C_4H_6$ and n-$C_4H_8$ exhibit strong interactions with $SO_4^{2-}$ pillars and pyridyl rings through multiple C-H•••O, C-H•••C, and C-H•••H interactions. Charge bias analysis reveals the charge shifts of positive H atoms in $C_4H_6$ and n-$C_4H_8$ to negative states, while O atoms in $SO_4^{2-}$ pillars shift towards positive potential. Furthermore, dynamic breakthrough experiments demonstrate the direct production of high-purity iso-$C_4H_8$ (>99.9%) from binary and ternary components mixtures.

## Results
### Structure characterization

The reaction of NiSO$_4$·6H$_2$O and DPDS in methanol solutions at room temperature yielded light blue powder of SOFOUR-DPDS-Ni with the chemical formula of Ni(DPDS)$_2$SO$_4$ (Fig. 1c, see methods for details). Note that the synthesis must be conducted under anhydrous conditions as H$_2$O molecules possess a significantly stronger coordination ability compared to $SO_4^{2-}$ anions, and may therefore occupy the coordination sites for Ni$^{2+}$ ions[35]. Despite numerous attempts, high-quality single crystals of SOFOUR-DPDS-Ni could not be obtained for single crystal X-ray diffraction analysis. The Rietveld refinements of powder X-ray diffraction (PXRD) data revealed that the as-synthesized SOFOUR-DPDS-Ni crystallizes in the orthorhombic crystal system with specific cell parameters of a = 10.5377, b = 14.2957, c = 19.8289 (Supplementary Fig. 2 and Supplementary Table 8). The well matched PXRD and simulated XRD patterns confirmed the high phase-purity of bulk SOFOUR-DPDS-Ni (Supplementary Fig. 3). Each Ni$^{2+}$ ion was coordinated with four pyridyl N atoms from four independent DPDS ligands in a distorted octahedral environment, forming one-dimensional chains of [Ni(DPDS)$_2$]$_n$ (Supplementary Fig. 4). These

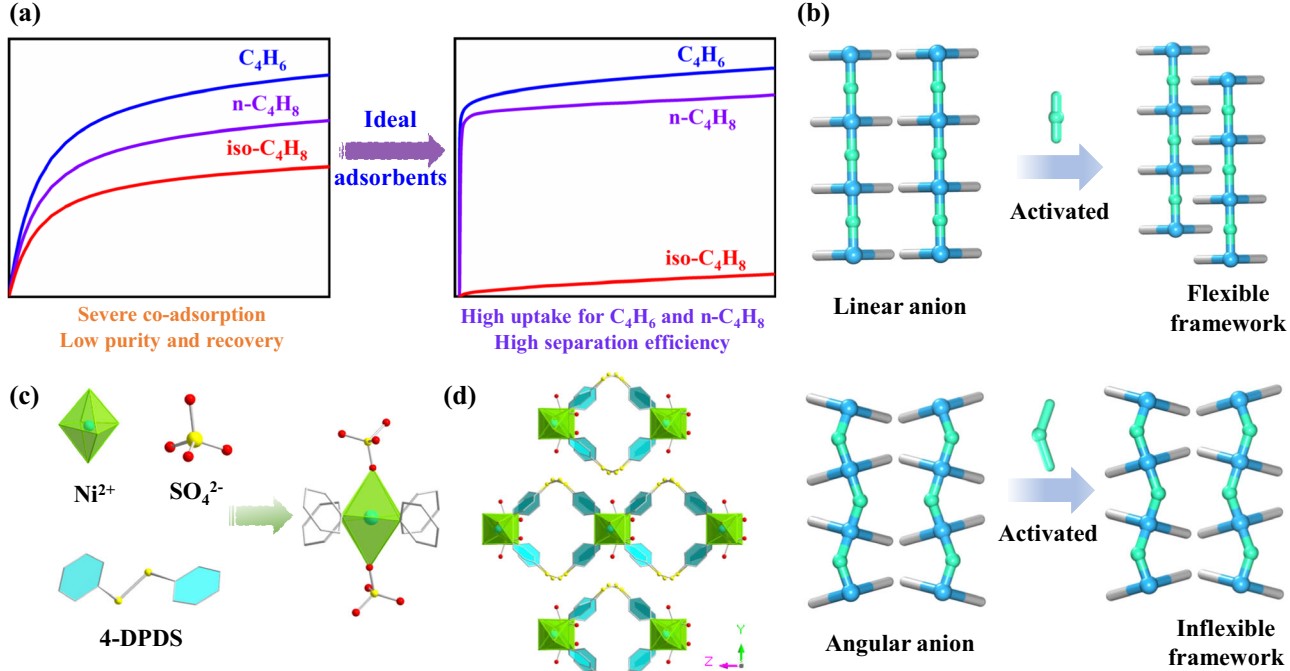

**Fig. 1 | Scheme and structure of SOFOUR-DPDS-Ni.** Schematic illustration of **a** adsorption behaviors on ideal adsorbents and **b** structure changes of adsorbents linked by linear and angular pillars after activation; **c** Building blocks (Ni$^{2+}$, $SO_4^{2-}$, and DPDS organic ligand) and local coordination environment of metal atoms; **d** SOFOUR-DPDS-Ni structure along the *X-axes*.

chains were further pillared by two $SO_4^{2-}$ anions in the axial direction, generating two-dimensional (2D) layers of [SOFOUR-DPDS-Ni]$_n$ with the *sql* topology. Adjacent 2D layers are assembled into three-dimensional structures via multiple π-π stacking interactions among pyridyl rings and p-π interactions between S atoms and pyridyl rings (Fig. 1d)[36]. The pore sizes of resulting intralayer and interlayer channels were measured to be $3.3 \times 4.5$ and $3.6 \times 3.7$ Å$^2$, respectively (Supplementary Fig. 5).

Thermogravimetric analysis (TGA) revealed the removal of guest molecules at 383 K, and demonstrated structural stability up to 473 K under N$_2$ atmosphere (Supplementary Fig. 6). The permanent porosity of activated SOFOUR-DPDS-Ni was probed by CO$_2$ gas sorption isotherms at 195 K, and the Brunauer-Emmett-Teller (BET) specific surface area was calculated to be 270 m$^2$ g$^{-1}$ with a total pore volume of 0.15 cm$^3$ g$^{-1}$ (Supplementary Fig. 7). Whereas, the N$_2$ adsorption in SOFOUR-DPDS-Ni was limited at 77 K. Based on the Horvath-Kawazoe model, the experimental pore size distributions exhibited a centered pore size of approximately 4.7 Å (Supplementary Fig. 8), which was consistent with the dimensions of intralayer cavities ($5.3 \times 4.5 \times 5.8$ Å$^3$) derived from the simulated crystal structure. The pore sizes of SOFOUR-DPDS-Ni were larger than the kinetic diameters of C$_4$H$_6$ (4.31 Å) and n-C$_4$H$_8$ (4.46 Å), yet smaller than that of iso-C$_4$H$_8$ (4.84 Å), suggesting the potential molecular sieving of iso-C$_4$H$_8$ from C$_4$H$_6$ and n-C$_4$H$_8$ counterparts. Additionally, the adsorption isotherms exhibited a high C$_2$H$_2$ uptake (2.87 mmol g$^{-1}$) and the moderate CO$_2$ uptake (1.36 mmol g$^{-1}$) at 298 K and 1.0 bar (Supplementary Fig. 9). The C$_2$H$_2$/CO$_2$ separation performance was considerably inferior compared to the molecular sieving effect on SO$_4^{2-}$-pillared SOFOUR-TEPE-Zn[31].

The PXRD pattern remained unchanged after activation, indicating the rigidity of SOFOUR-DPDS-Ni (Supplementary Fig. 3). Due to the presence of angular SO$_4^{2-}$ pillars connecting adjacent 2D layers, the obtained unparallel stacking effectively prevented layer-to-layer sliding after removal of guest molecules (Fig. 1b). Additionally, no gate-opening or step-wise adsorption behavior was observed in the

measured adsorption isotherms of C$_4$H$_6$ and n-C$_4$H$_8$ at different temperatures (Supplementary Fig. 10). The excellent chemical stability was demonstrated by its ability to withstand soaking in various organic solvents for one week, hot water at 60 °C for 2 h, and exposure to air for 13 months (Supplementary Fig. 11). The ultimate elemental analysis demonstrated that the element composition of each element corresponded well with the theoretical formula of SOFOUR-DPDS-Ni (C$_{20}$H$_{16}$N$_4$O$_4$S$_5$Ni, Supplementary Table 1). For instance, the measured content ratio of N/S (0.39) was closely matched to the theoretical value (0.35). The scanning electron microscopy (SEM) image revealed a block morphology of SOFOUR-DPDS-Ni (Supplementary Fig. 12). Fourier transform infrared spectroscopy (FT-IR) spectra exhibited characteristic peaks corresponding to stretching vibrations for Ni-O at 493.2 cm$^{-1}$ and S-O in SO$_4^{2-}$ at 1058.3 cm$^{-1}$. Detailed discussions regarding FT-IR results are presented below Supplementary Fig. 13.

## Gas adsorption and separation behaviors

Single-component gas adsorption isotherms of C$_4$H$_6$, n-C$_4$H$_8$, and iso-C$_4$H$_8$ were collected on SOFOUR-DPDS-Ni (Fig. 2a and Supplementary Fig. 10). The adsorption capacity for C$_4$H$_6$ and n-C$_4$H$_8$ was measured to be 1.68 and 1.48 mmol g$^{-1}$ at 298 K and 1.0 bar, respectively. In contrast, iso-C$_4$H$_8$ was almost completely excluded with a negligible uptake of only 0.17 mmol g$^{-1}$. The kinetic adsorption curve disclosed that C$_4$H$_6$ reached adsorption equilibrium within 5 min, while n-C$_4$H$_8$ required a longer time of ~30 min to reach equilibrium (Fig. 2b). Whereas, iso-C$_4$H$_8$ exhibited negligible adsorption even after an extended exposure of 35 min. The kinetic adsorption capacity for C$_4$H$_6$ and n-C$_4$H$_8$ was determined to be 1.47 and 1.17 mmol g$^{-1}$ at 0.5 bar, which were comparable to their static adsorption uptakes. The diffusion time constant (D/r$^2$) was calculated to be $1.29 \times 10^{-3}$ s$^{-1}$ for C$_4$H$_6$ and $2.72 \times 10^{-4}$ s$^{-1}$ for n-C$_4$H$_8$, suggesting a faster diffusion rate for C$_4$H$_6$ in SOFOUR-DPDS-Ni (Supplementary Fig. 14). Furthermore, molecular dynamic (MD) simulations demonstrated that the diffusion coefficient of C$_4$H$_6$

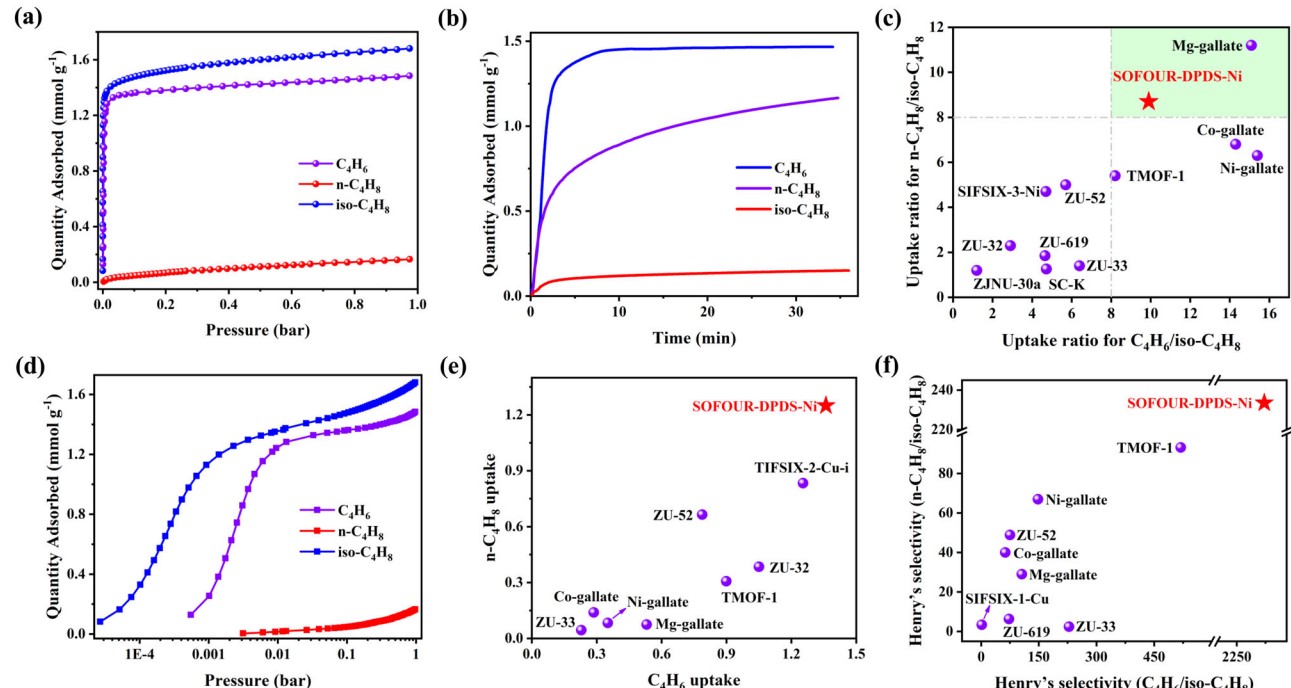

**Fig. 2 | C$_4$H$_6$, n-C$_4$H$_8$, and iso-C$_4$H$_8$ sorption in SOFOUR-DPDS-Ni. a** Pure-component isotherms at 298 K and **b** kinetic adsorption curves at 298 K and 0.5 bar for C$_4$H$_6$, n-C$_4$H$_8$, and iso-C$_4$H$_8$; **c** Comparison of uptake ratios of C$_4$H$_6$/iso-C$_4$H$_8$ and n-C$_4$H$_8$/iso-C$_4$H$_8$ at 1.0 bar; **d** Adsorption isotherms in logarithmic form for C$_4$H$_6$, n-C$_4$H$_8$, and iso-C$_4$H$_8$ at 298 K; **e** Comparison of uptakes of C$_4$H$_6$ and n-C$_4$H$_8$ at 0.01 bar; **f** Comparison of Henry's selectivity for C$_4$H$_6$/iso-C$_4$H$_8$ and n-C$_4$H$_8$/iso-C$_4$H$_8$.

$(4.82 \times 10^{-11})$ was faster than that of n-C$_4$H$_8$ $(1.02 \times 10^{-11})$ at 298 K (Supplementary Figs. 15 and 16). The kinetic separation selectivity of 4.7 for C$_4$H$_6$/n-C$_4$H$_8$ indicated a limited kinetic contribution to the apparent selectivity. Benefiting from the molecular sieving effect of iso-C$_4$H$_8$, uptake ratios were utilized as an intuitive measure of selectivity for separating C$_4$ olefins. At 1.0 bar and 298 K, the uptake ratios of C$_4$H$_6$/iso-C$_4$H$_8$ and n-C$_4$H$_8$/iso-C$_4$H$_8$ on SOFOUR-DPDS-Ni were calculated to be 9.9 and 8.7, surpassing most top-ranking adsorbents including TMOF-1 (8.2 and 5.4)[1], ZU-52 (5.7 and 5.0)[24], and SIFSIX-3-Ni (4.7 and 4.7)[24] (Fig. 2c and Supplementary Table 6). In comparison, we measured the adsorption isotherms of C$_4$ olefins on SOFOUR-1-Zn and SOFOUR-TEPE-Zn at 298 K (Supplementary Fig. 17). SOFOUR-1-Zn adsorbed comparable uptakes for n-C$_4$H$_8$ (0.72 mmol g$^{-1}$) and iso-C$_4$H$_8$ (0.55 mmol g$^{-1}$). Meanwhile, negligible n-C$_4$H$_8$ and iso-C$_4$H$_8$ uptakes (<0.13 mmol g$^{-1}$) were observed on SOFOUR-TEPE-Zn. These inferior performances on SO$_4$$^{2-}$-pillared adsorbents further highlighted the advantages of pore environments of SOFOUR-DPDS-Ni.

Remarkably, SOFOUR-DPDS-Ni exhibited high adsorption capacity of 1.36 mmol g$^{-1}$ for C$_4$H$_6$ and 1.25 mmol g$^{-1}$ for n-C$_4$H$_8$ at a low pressure of 0.01 bar (Fig. 2d). At even lower pressure of 0.001 bar (1000 ppm), the C$_4$H$_6$ uptake unprecedently reached 1.15 mmol g$^{-1}$, suggesting its potential removal of trace C$_4$H$_6$ from C$_4$ olefin mixtures. The adsorption affinities for C$_4$H$_6$ and n-C$_4$H$_8$ were further confirmed by the isosteric heats of adsorption ($Q_{st}$) using the Clausius-Clapeyron equation[37] (Supplementary Fig. 18). Specifically, the $Q_{st}$ was calculated to be 77 kJ mol$^{-1}$ and 38 kJ mol$^{-1}$ for C$_4$H$_6$ and n-C$_4$H$_8$ at near zero coverage, respectively (Supplementary Fig. 19). The high uptakes of both C$_4$H$_6$ and n-C$_4$H$_8$ established a new

benchmark for iso-C$_4$H$_8$ separation, outperforming most adsorbents including ZU-32[24], TMOF-1[1], and Ni-gallate[38] (Fig. 2e). Moreover, the Henry's constant of C$_4$H$_6$ and n-C$_4$H$_8$ was calculated to be 50.87 and 5.11 mmol g$^{-1}$ kPa$^{-1}$, which were significantly higher than that of iso-C$_4$H$_8$ (0.022 mmol g$^{-1}$ kPa$^{-1}$). As a result, the Henry's selectivity for C$_4$H$_6$/iso-C$_4$H$_8$ and n-C$_4$H$_8$/iso-C$_4$H$_8$ on SOFOUR-DPDS-Ni reached high values of 2321.8 and 233.5 at 298 K, respectively. As shown in Fig. 2f, these values were much higher than other reported adsorbents such as TMOF-1 (519.2 and 93.2)[1], ZU-33 (228.7 and 2.4)[24], and ZU-52 (75.2 and 48.9)[24] (Supplementary Table 7).

## Simulation studies

Grand Canonical Monte Carlo (GCMC) and first-principles dispersion-corrected density function theory (DFT-D) simulations were employed to elucidate the adsorption mechanism of C$_4$ olefins on SOFOUR-DPDS-Ni. The distribution densities of C$_4$H$_6$ and n-C$_4$H$_8$ were investigated via GCMC simulations at 0.01 bar and 1.0 bar, both gas molecules were adsorbed in the intralayer spaces near SO$_4$$^{2-}$ pillars (Fig. 3a, b). It is noteworthy that the distribution densities of C$_4$H$_6$ and n-C$_4$H$_8$ were nearly identical at pressures of 0.01 and 1.0 bar (Supplementary Figs. 21 and 22), which aligned with their early adsorption saturation at low pressures in SOFOUR-DPDS-Ni. Moreover, four C$_4$H$_6$ or n-C$_4$H$_8$ molecules were adsorbed per unit cell, their uptakes were accordingly calculated to be 1.68 mmol g$^{-1}$ at 1.0 bar, which were closely matched to their experimental uptakes.

DFT-D simulations revealed two favorable adsorption sites for C$_4$H$_6$ in the interlayer cavities of SOFOUR-DPDS-Ni, which were captured by SO$_4$$^{2-}$ anions through multiple C-H•••O interactions with

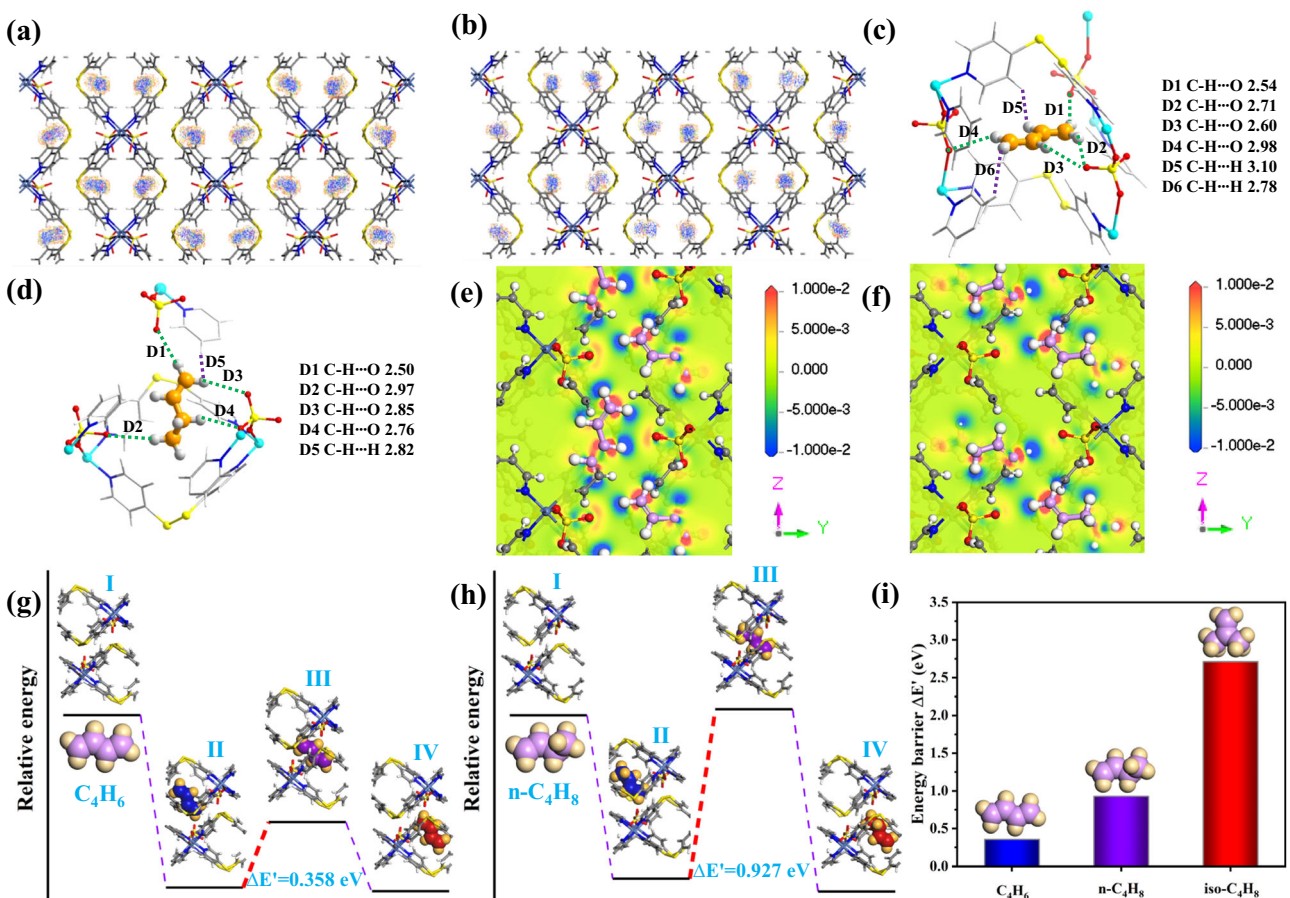

**Fig. 3 | GCMC simulated and DFT-D calculated results in SOFOUR-DPDS-Ni.** Distribution density for **a** C$_4$H$_6$ and **b** n-C$_4$H$_8$ in SOFOUR-DPDS-Ni at 1.0 bar; Adsorption binding site for **c** C$_4$H$_6$ and **d** n-C$_4$H$_8$ in SOFOUR-DPDS-Ni; Charge density difference plots of **e** C$_4$H$_6$-loaded and **f** n-C$_4$H$_8$-loaded structure; Illustration of diffusion pathway and corresponding energy levels **g** C$_4$H$_6$ and **h** n-C$_4$H$_8$; **i** Comparison of diffusion energy barriers for C$_4$H$_6$, n-C$_4$H$_8$, and iso-C$_4$H$_8$.

distances of 2.41–2.98 Å and pyridine rings through multiple van der Walls interactions of C-H•••C and C-H•••H with distances of 2.77–3.10 Å (Fig. 3c and Supplementary Fig. 23a). The calculated static binding energy ($\Delta E$) for Site I and Site II were estimated to be 78.3 and 80.6 kJ mol$^{-1}$, respectively (Supplementary Fig. 24). Due to the conformational changes of n-$C_4H_8$, four favorable adsorption sites for n-$C_4H_8$ were identified, while the resulting interactions were comparable to those of $C_4H_6$. Specifically, n-$C_4H_8$ molecules strongly interacted with $SO_4^{2-}$ anions through multiple C-H•••O interactions with distances of 2.43–2.98 Å and pyridine rings through multiple van der Walls interactions of C-H•••C and C-H•••H with distances of 2.24–2.95 Å (Fig. 3d and Supplementary Fig. 23b–d). The calculated $\Delta E$ values for the four binding sites were 72.8, 58.3, 64.6, and 71.2 kJ mol$^{-1}$, respectively (Supplementary Fig. 24). Furthermore, gas-loaded structures were subjected to charge transfer analysis, with the blue and yellow surfaces indicating charge accumulation and depletion, respectively. Upon adsorption, the initial positively charged H atoms in $C_4H_6$ and n-$C_4H_8$ were shifted to strong negative potentials, while O atoms in $SO_4^{2-}$ pillars shifted to positive potential (Fig. 3e, f). In addition, charge transfer also occurred between the adsorbed guests and the pyridine rings (Supplementary Fig. 25). Therefore, the strong interaction forces between [SOFOUR-DPDS-Ni]$_n$ layers (i.e., $SO_4^{2-}$ pillars and pyridyl rings) and adsorbed guests (i.e., $C_4H_6$ and n-$C_4H_8$) stabilized the framework of SOFOUR-DPDS-Ni, thereby endowing it with rigidity during $C_4$ adsorptions.

The energy levels associated with the diffusions processes of $C_4H_6$, n-$C_4H_8$, and iso-$C_4H_8$ into SOFOUR-DPDS-Ni framework were further investigated. The rate-determining step was attributed to the largest energy input required for the energy barrier of transition-states (TS) between surface adsorption TS (II) and diffusion into pores TS (III). The diffusion energy barrier of $C_4H_6$ in SOFOUR-DPDS-Ni was determined to be 0.358 eV, indicating its facile transport through interlayer channels (Fig. 3g). The diffusion energy barrier increased to 0.927 eV for n-$C_4H_8$, attributing to its larger steric hindrance caused by the methyl group (Fig. 3h). In contrast, the overwhelming diffusion energy barrier for iso-$C_4H_8$ (2.712 eV) suggested inhibited diffusions thus causing the negligible adsorption amount (Fig. 3i and Supplementary Fig. 26).

## Transient breakthrough experiments

Transient breakthrough experiments were conducted to evaluate the practical separation performances of SOFOUR-DPDS-Ni for $C_4$ mixtures[39], which exhibit varying component proportions due to diverse streams from different stream cracking processes (Supplementary Fig. 27). Here, $C_4$ olefin gas-mixtures of $C_4H_6$/iso-$C_4H_8$ (50/50, $v/v$), n-$C_4H_8$/iso-$C_4H_8$ (50/50, $v/v$), and $C_4H_6$/n-$C_4H_8$/iso-$C_4H_8$ (50/15/35, $v/v/v$) were selected as representative gas-mixtures for simulating the actual separation process. For $C_4H_6$/iso-$C_4H_8$ mixture with a flow rate of 1.0 mL min$^{-1}$, iso-$C_4H_8$ was eluted first at the outlet of packed column, while $C_4H_6$ was not detected until 57.6 min (Fig. 4a). The kinetic adsorption capacity of $C_4H_6$ on SOFOUR-DPDS-Ni was calculated to be 1.48 mmol g$^{-1}$, which closely approximated its static uptake of 1.59 mmol g$^{-1}$ at 0.5 bar. High-purity iso-$C_4H_8$ (>99.9%) could be collected at the outlet with a productivity of 1.20 mmol g$^{-1}$. Furthermore, high-purity $C_4H_6$ (>99.0%) could be collected from 14.4 min during the desorption process with a productivity of 1.17 mmol g$^{-1}$

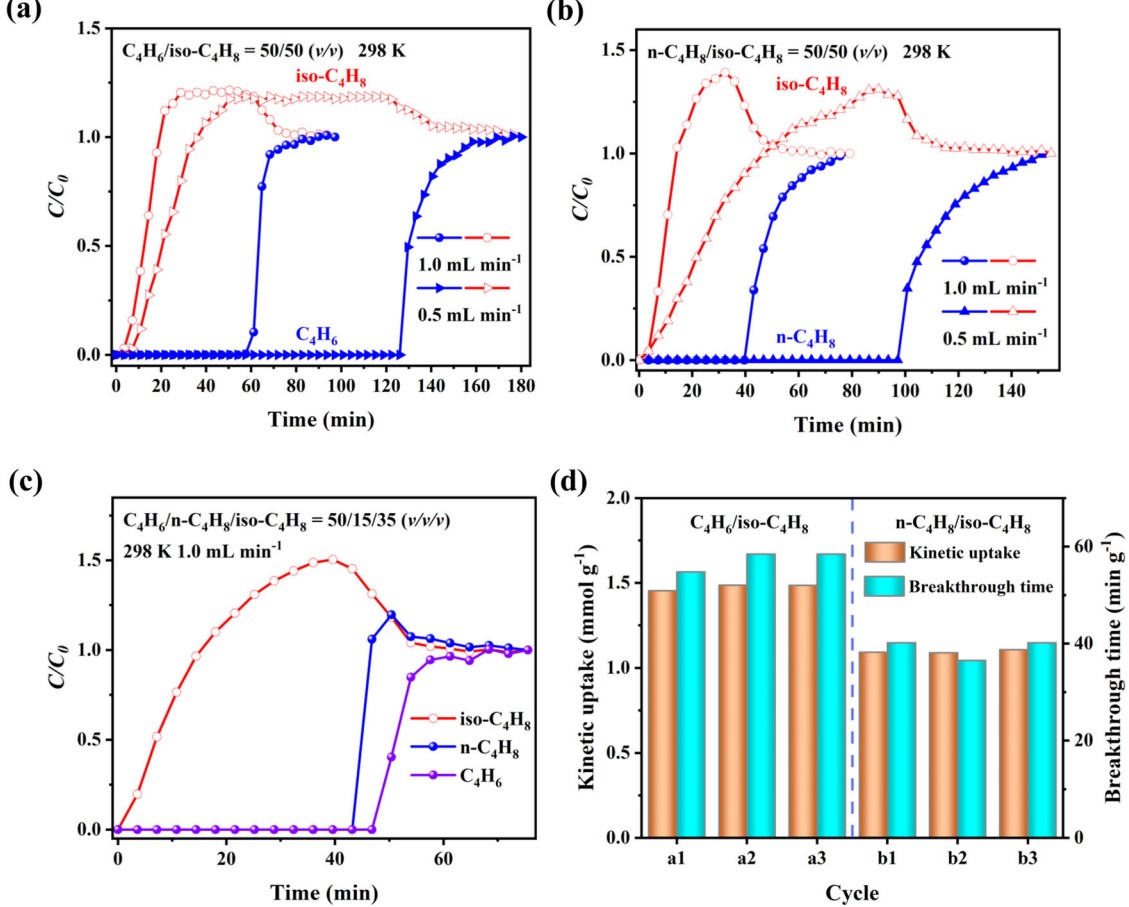

**Fig. 4 | $C_4H_6$, n-$C_4H_8$, and iso-$C_4H_8$ separation performances.** Experimental column breakthrough curves for **a** $C_4H_6$/iso-$C_4H_8$ (50/50, $v/v$), **b** n-$C_4H_8$/iso-$C_4H_8$ (50/50, $v/v$), and **c** $C_4H_6$/n-$C_4H_8$/iso-$C_4H_8$ (50/15/35, $v/v/v$); **d** Dynamic uptakes for three cycling breakthrough tests for $C_4H_6$/iso-$C_4H_8$ (a1-a3) and n-$C_4H_8$/iso-$C_4H_8$ (b1-b3) at 1.0 mL min$^{-1}$.

using He purge at 5.0 mL min$^{-1}$ and 343 K (Supplementary Fig. 28). The efficient separation of $C_4H_6$/iso-$C_4H_8$ could also be achieved at a slower flow rate of 0.5 mL min$^{-1}$ with a comparable kinetic $C_4H_6$ uptake of 1.51 mmol g$^{-1}$. Similarly, for n-$C_4H_8$/iso-$C_4H_8$ gas-mixture, iso-$C_4H_8$ immediately broke through the column, followed by n-$C_4H_8$ at 39.6 min at 1.0 mL min$^{-1}$ (Fig. 4b). The dynamic uptake of n-$C_4H_8$ was determined to be 1.09 mmol g$^{-1}$, which was 76.8% of its static adsorption uptake (1.42 mmol g$^{-1}$) at 0.5 bar. High-purity iso-$C_4H_8$ (>99.9%) effluent could be obtained with a productivity of 0.85 mmol g$^{-1}$. While high-purity n-$C_4H_8$ (>99.0%) could also be collected during the desorption process with a productivity of 0.74 mmol g$^{-1}$ (Supplementary Fig. 29). The kinetic uptake of n-$C_4H_8$ increased to 1.28 mmol g$^{-1}$ (90.1% of static uptake) at 0.5 mL min$^{-1}$, indicating the slower diffusion rate of n-$C_4H_8$ compared to that of $C_4H_6$.

Moreover, efficient separation of a ternary gas-mixture containing $C_4H_6$/n-$C_4H_8$/iso-$C_4H_8$ (50/15/35, v/v/v) at 1.0 mL min$^{-1}$ was successfully achieved using SOFOUR-DPDS-Ni as well. Figure 4c depicted that iso-$C_4H_8$ was immediately detected, while $C_4H_6$ and n-$C_4H_8$ were retained in the column for 46.8 min and 43.2 min, respectively. Their competitive adsorptions will simultaneously occupy the available adsorption sites and inevitably influence their adsorption uptakes. High-purity iso-$C_4H_8$ (>99.9%) could be directly collected with a productivity of 0.71 mmol g$^{-1}$ during a time period of 43.2 min. The reusability of SOFOUR-DPDS-Ni was validated through three successive breakthrough cycles using binary and ternary $C_4$ gas-mixtures (Supplementary Figs. 30–35). The cycling breakthrough points and curve shapes were almost overlapped, indicating consistent separation performance. Furthermore, the kinetic adsorption capacity for $C_4H_6$ and n-$C_4H_8$ also remained consistent at 0.5 and 1.0 mL min$^{-1}$ throughout the cycling process (Fig. 4d and Supplementary Fig. 36). Furthermore, the PXRD patterns of SOFOUR-DPDS-Ni remained change after cycled adsorption tests and breakthrough experiments, thereby indicating its exceptional structural stability (Supplementary Fig. 39).

## Discussion

In summary, we successfully synthesized the sulfate-pillared adsorbent, SOFOUR-DPDS-Ni, for efficient separation of iso-$C_4H_8$ from $C_4$ olefin mixtures. Simultaneous high uptakes for $C_4H_6$ and n-$C_4H_8$ of 1.68 and 1.48 mmol g$^{-1}$ at 298 K and 1.0 bar were achieved on SOFOUR-DPDS-Ni, while exhibiting size-sieving effect for iso-$C_4H_8$. At low pressures of 0.01 bar, SOFOUR-DPDS-Ni exhibited high adsorption capacity of 1.36 mmol g$^{-1}$ for $C_4H_6$ and 1.25 mmol g$^{-1}$ for n-$C_4H_8$. Notably, at an ultralow pressure of 0.001 bar (1000 ppm), the unprecedented $C_4H_6$ adsorption capacity of 1.15 mmol g$^{-1}$ indicated the potential for trace $C_4H_6$ removal. Therefore, the benchmark uptake ratio and Henry's selectivity for $C_4H_6$/iso-$C_4H_8$ (9.9 and 2321.8) and n-$C_4H_8$/iso-$C_4H_8$ (8.7 and 233.5) surpassed most top-ranking adsorbents. GCMC and DFT-D simulations demonstrated the adsorption sites and separation mechanisms. Breakthrough experiments confirmed the practical iso-$C_4H_8$ separation performances of SOFOUR-DPDS-Ni from binary and ternary $C_4$ olefins mixtures. The high-purity iso-$C_4H_8$ (99.9%) could be directly collected, meanwhile high-purity $C_4H_6$ (99.0%) and n-$C_4H_8$ (99.0%) could also be obtained during desorption processes.

## Methods
### Materials

All reagents and solvents were obtained from commercial sources and used without further purification. Nickel sulfate hexahydrate (NiSO$_4$·6H$_2$O, 99.0%, Aladdin), 4,4'-dipyridyl disulfide ($C_{10}H_8N_2S_2$, 98.0%, Xiya Reagent), and methanol (CH$_4$O, anhydrous, 99.9%, Aladdin). 1,3-butadiene ($C_4H_6$, 99.9%), n-butene (n-$C_4H_8$, 99.9%), iso-butene (iso-$C_4H_8$, 99.9%), N$_2$ (99.999%), He (99.999%), and mixed gas-mixtures of $C_4H_6$/iso-$C_4H_8$ (50/50, v/v), n-$C_4H_8$/iso-$C_4H_8$ (50/50, v/v),

and $C_4H_6$/n-$C_4H_8$/iso-$C_4H_8$ (50/15/35, v/v/v) were purchased from Nanchang Jiangzhu Gas Co., Ltd (China).

### Synthesis of SOFOUR-DPDS-Ni
The material was designated as SOFOUR-DPDS-Ni, with a chemical formula of Ni(DPDS)$_2$SO$_4$. NiSO$_4$·6H$_2$O (0.2 mmol, 0.0526 g) was added to a solution of 4-DPDS (0.4 mmol, 0.0881 g) in 20 mL MeOH and stirred at room temperature for 24 h. SOFOUR-DPDS-Ni was obtained as a light blue powder and washed with 100 mL MeOH, followed by drying for 6 h at room temperature.

### Details for Rietveld refinement
We applied the EXPO2014 software to conduct the Rietveld refinement, the 2θ of 5-60° was used for the refinement. Chebyshev (Background Function) and Pseudo-Voigt (Peak Shape Functions) were applied to refine the structure until the $R_{wp}$ value converged and the overlay of the observed with refined profiles showed good agreement. Unit cell parameters and fitting reliability are listed in Supplementary Table 8, and we have deposited the CIF in the CCDC database with an identifier number of 2260840.

### Characterizations
Powder X-ray diffraction (PXRD) analysis of powder samples was carried out on a PANalytical Empyrean Series 2 diffractometer with Cu Kα radiation ((λ = 1.540598 Å), which operated at 40 kV, 40 mA and a scan speed of 0.0167°, a scan time of 15 s per step and 2θ ranging from 5 to 60° at room temperature. The thermogravimetric analysis (TGA) data were obtained on a NETZSCH Thermogravimetric Analyzer (STA2500) from 25 to 800 °C with a heating rate of 20 °C min$^{-1}$ under an N$_2$ atmosphere. The contents of C, H, N, and S elements were determined by an Elementar Vario MICRO elemental analyzer with CHNS measurement mode. The SEM images were recorded on a Thermo Scientific Apreo 2C scanning electron microscope with an accelerated voltage of 10 kV. The Fourier transfer infrared spectroscopy (FT-IR) was tested by NICOLET FT-IR spectrometer (iS50 FI-IR), and the resolution was 4 cm$^{-1}$, the number of scans was 32, and the test wave number was 400–4000 cm$^{-1}$.

### Gas adsorption measurements
Single-component isotherms of $C_4H_6$, n-$C_4H_8$, and iso-$C_4H_8$ were measured up to 1 bar at 283 and 298 K on Micromeritics 3Flex adsorption apparatus (Micromeritics Instruments, USA). The kinetic adsorptions of $C_4H_6$, n-$C_4H_8$, and iso-$C_4H_8$ were obtained on Intelligent Gravimetric Analyzer (IGA−100, HIDEN), and the pressure rise rate is 200 mbar min$^{-1}$. About 100 mg powder samples were evacuated under high vacuum (<5 μm of Hg) at 70 °C for 12 h before adsorption measurement, and the free space of the system was measured by using helium gas. Liquid nitrogen and dry ice were used for adsorption isotherms at 77 K and 195 K, the pore size distribution was calculated based on CO$_2$ adsorption isotherms at 195 K.

### Transient breakthrough experiments
The breakthrough experiments were implemented in a stainless-steel column (4.6 mm inner diameter ×200 mm) manually packed with the weight of 0.9853 g activated SOFOUR-DPDS-Ni. The column was first purged with a He flow (10 mL min$^{-1}$) at room temperature for 10 h before breakthrough tests. The mixtures of $C_4H_6$/iso-$C_4H_8$ (50/50, v/v), n-$C_4H_8$/iso-$C_4H_8$ (50/50, v/v), and $C_4H_6$/n-$C_4H_8$/iso-$C_4H_8$ (50/15/35, v/v/v) was conducted at a flow rate of 0.5 and 1.0 mL min$^{-1}$, respectively. The outlet gas from the column was monitored using gas chromatography (Panna A91 Plus GC) for continuous sampling gas analysis, and an attached mass flow controller (Seven Star, MC-2SCCM-D, D07 series) was used to control the gas flow rate. After the breakthrough tests, the columns packed with samples were regenerated by purging He gas of 10 mL min$^{-1}$ at 70 °C for 8 h. For the $C_4$ mixtures of

$C_4H_6$/iso-$C_4H_8$ (50/50, $v/v$) and n-$C_4H_8$/iso-$C_4H_8$ (50/50, $v/v$), desorption process was carried out in order to obtain high-purity $C_4H_6$ and n-$C_4H_8$ after the breakthrough experiment. The desorption process was conducted under He gas flow rate of 5.0 mL min$^{-1}$ at 70 °C.

## Structural stability tests

The activated samples of about 100 mg were placed in 20 mL vials containing 10 mL different solvents for 7 days, treated in hot water of 60 °C for 2 h, exposed air for thirteen months, respectively. The treated samples were washed with 100 mL MeOH and dried at room temperatures, and then characterized by PXRD measurements to determine whether the sample retains structural integrity.

## Grand Canonical Monte Carlo (GCMC) calculations

All the GCMC simulations were performed in Materials Studio package. The framework, $C_4H_6$, and n-$C_4H_8$ were considered to be rigid during the simulation. The charges for atoms of the SOFOUR-DPDS-Ni and gas components were derived from the Mulliken method. The simulations adopted the fixed pressure task, Metropolis method in sorption module, and the universal force field (UFF). The interaction energy between the adsorbed molecules and the framework was computed through Lennard-Jones 6–12 (LJ) potentials. The cutoff radius was chosen 15.5 Å and the electrostatic interactions were handled using the Ewald summation method. The loading steps and the equilibration steps were $1 \times 10^7$, the production steps were $1 \times 10^7$.

## Density Functional Theory calculations

First-principles density functional theory (DFT) calculations were performed using the Materials Studio's CASTEP code. All calculations were conducted under the generalized gradient approximation (GGA) with Perdew–Burke–Ernzerhof (PBE). A semiempirical addition of dispersive forces to conventional DFT was included in the calculation to account for van der Waals interactions. The total energy coverage within 0.01 meV atom$^{-1}$. The optimization process commenced with refine structures of the synthesized materials. The charge transfer analysis on gas-loaded structures was calculated using "Electron density difference" in properties of CASTEP module. Single point energy calculations using Dmol$^3$ module. To obtain the binding energy, an isolated gas molecule placed in a cell unit (with the same cell dimensions as the MOF crystal). The static binding energy was calculated by the equation:

$$\Delta E = E(\text{gas}) + E(\text{adsorbent}) - E(\text{adsorbent} + \text{gas}) \quad (1)$$

## The energy barrier calculation method

The energy barrier calculations were carried out using the Dmol$^3$ module in Materials Studio. The unit cells were optimized until the force acting between atoms was below 0.002 Ha Å$^{-1}$ with SCF convergence of $10^{-6}$. The Global orbital cutoff was 5.2 Å. The diffusion of guest molecules was studied by determining the transition state energies using the climbing nudged elastic band (cNEB) method. Firstly, the surface model and the host structure would be optimized using the refine structures as initial geometries with full structural relaxation. The isolated guest molecules ($C_4H_6$, n-$C_4H_8$, and iso-$C_4H_8$) were placed in unit cell and relaxed as references. Next, the guest molecules were introduced onto the host surface and different locations in the channel pore of the host structure, respectively, followed by a full structural relaxation. Then the optimized configurations of the lowest energy were utilized for the subsequent analysis and calculation. The transition state search calculations were used to capture the transition states with guest transport between the two energy minimum configurations from the host surface to channels.

The energy barrier was determined using the following:

$$\Delta E' = E(\text{Transition State}) - E(\text{Initial State}) \quad (2)$$

where $E$(Transition State) is the transition energy, $E$(Initial State) is the energy of the optimized host-guest structure where guests were introduced onto the host surface.

## Calculation of selectivity

The single-component adsorption isotherms of $C_4$ hydrocarbons were correlated by the Langmuir model at low pressure of 0–5 kPa. The Langmuir model was defined as:

$$q = \frac{q_m bp}{1 + bp} \quad (3)$$

where, $q$ is the adsorbed amount of the pure component i (mmol g$^{-1}$), $p$ is the pressure of the bulk gas at equilibrium (kPa), $q_m$ is the saturated adsorption capacities (mmol g$^{-1}$), $b$ is the affinity parameters of the pure component (kPa$^{-1}$).

To estimate the separation selectivity of SOFOUR-DPDS-Ni, Henry's selectivity ($\alpha_{ij}$) was developed and applied, which reflected separation selectivity at low pressure about 0–5 kPa. The Henry's selectivity ($\alpha_{ij}$) based on equilibrium alone can be calculate from the ratio of Henry's constants, $H = q_m \times b$. The selectivity was defined by the following equation:

$$\alpha_{ij} = \frac{H_i}{H_j} = \frac{q_{mi} b_i}{q_{mj} b_j} \quad (4)$$

## Isosteric heat of adsorption

The experiment isosteric heat of adsorption for $C_4H_6$ and n-$C_4H_8$ were calculated using the data at 283 K and 298 K, which was calculated by the Clausius–Clapeyron equation and was defined as:

$$Q_{st} = -RT^2 \left( \frac{\partial \ln P}{\partial T} \right) \quad (5)$$

where $Q_{st}$ (kJ mol$^{-1}$) represents the adsorption heat of gas molecular, $P$ (mmHg) and $T$ (K) represent the pressure and temperature, respectively, and $R$ is the universal gas constant. Here, the adsorption heat of each component was determined precisely according to the virial fitting parameters of single-component adsorption isotherms measured at 283 and 298 K, which was calculated as follows:

$$\ln P = \ln N + \frac{1}{T} \sum_{i=0}^{m} a_i N^i + \sum_{j=0}^{n} b_j N^j \quad (6)$$

$$Q_{st} = -R \sum_{i=0}^{m} a_i N^i \quad (7)$$

where the $N$ (mg g$^{-1}$) is the adsorption amount, and $m$ and $n$ determine the number of items required to precisely fit the adsorption isotherms.

## Calculation of kinetic adsorption

The diffusional time constants ($D'$, $D/r^2$) were calculated by the short-time solution of the diffusion equation assuming a step change in the gas-phase concentration, clean beds initially and micropore diffusion control:

$$\frac{M_t}{M_e} = \frac{6}{\sqrt{\pi}} \cdot \sqrt{\frac{D}{r^2} \cdot t} \quad (8)$$

where $t$ (s) is the time, $M_t$ (mmol g$^{-1}$) is gas uptake at time $t$, $M_e$ (mmol g$^{-1}$) is the gas uptake at equilibrium, $D$ (m$^2$ s$^{-1}$) is the diffusivity and $r$ (m) is the radius of the equivalent spherical particle. The slopes of $M_t/M_e$ versus $t^{1/2}$ are derived from the fitting of the plots at 0.5 bar and 298 K, and the pressure rise rate is 200 mbar min$^{-1}$.

## Data availability

All data supporting the finding of this study are available within this article and its Supplementary Information. Crystallographic data for the structure in this article have been deposited at the Cambridge Crystallographic Date Center under deposition nos. CCDC 2260840 (SOFOUR-DPDS-Ni). Correspondence and requests for materials should be addressed to J.W.

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

## Acknowledgements

This study was supported by the National Natural Science Foundation of China (No. 22322807, 22168023, 22308142, and 22268029) and the Natural Science Foundation of Jiangxi Province (No. 20224ACB204003).

## Author contributions

J.L. and J.W. conceived the project, designed the research, and co-wrote the manuscript. J.L. carried out the materials synthesis, adsorption experiments, transient breakthrough measurement, and computational simulations. H.X. carried out computational simulations. H.S. conducted the IAST and $Q_{st}$ calculations. X.L. and Y.P. conducted dynamic adsorption experiments. L.W. and P.W. collected the PXRD data. Z.W.Z. and Z.D. conducted stability testing. Z.Y.Z., J.C. and S.C. validated and visualized the results. Z.L.Z., S.D., and J.W. contributed to the discussion of the results and commented on the manuscript.

## Competing interests

The authors declare no competing interests.
