## [Peer Review File · Nature Communications]

Reviewer #1 (Remarks to author)

In this work, Wang and co-workers reported a novel sulfate-pillared adsorbent, SOFOUR-DPDS-Ni, for efficient separation of *iso*-C₄H₈ from both binary and ternary C₄ olefin mixtures. The suitable pore sizes and shapes facilitate the selective adsorption of C₄H₆ and *n*-C₄H₈ over *iso*-C₄H₈, achieving remarkable Henry's selectivity for C₄H₆/*iso*-C₄H₈ and *n*-C₄H₈/*iso*-C₄H₈. Breakthrough experiments confirmed the feasibility of producing high-purity *iso*-C₄H₈ directly, and obtaining high-purity C₄H₆ and *n*-C₄H₈ during desorption processes. This manuscript is well-organized, and the structure of adsorbent is interesting with excellent performance that provide some useful clues for designing high performance porous materials for challenging gas separation. I suggest it be published in *Nature Communications* after the following revisions.

1. More characterizations of SOFOUR-DPDS-Ni, such as EA, FT-IR, and SEM, are suggested.
2. The kinetic curve of C₄H₆ in Fig. S12 is not well-fitted, which may affect the accuracy of the results.
3. The crystallinity and integrity of the SOFOUR-DPDS-Ni should be checked after adsorption tests and cycling breakthrough experiments.
4. Detailed information for calculating the productivity of *iso*-C₄H₈ and the dynamic adsorption capacity of C₄H₆ and *n*-C₄H₈ should be provided.
5. The structure of *iso*-C₄H₈ molecule in Fig. 3i must be corrected.
6. The values of R_p and R_{wp} in Fig. S2 are missing.
7. Figs 2d and S9 are duplicates, please check.
8. The article should be formatted by the requirements of *Nat. Commun.*
9. The description of comparing the pore size of SOFOUR-DPDS-Ni with C₄ olefins in Lines 121 and 122 is wrong.
10. Please pay attention to the layout of Tables in the Supplementary Information.

REVIEWER COMMENTS

Reviewer #2 (Remarks to the Author):

This paper presents an innovative reticular chemistry modification applied to SOFOUR-1-Zn and SOFOUR-TEPE-Zn, which were originally investigated for C₂H₂/CO₂ separation. This modification results in the creation of a novel material termed SOFOUR-DPDS-Ni, now studied for C₄H₆/Iso-C₄H₈ and n-C₄H₈/iso-C₄H₈ separation. The discovery of SOFOUR-DPDS-Ni is undoubtedly noteworthy. However, it's worth noting that the transition to higher-density sulfate-pillared materials, or materials with equal density, for testing C₄ isomers isn't entirely novel, as it resonates with prior publications within this field. Examples of these references include:

Chemical Engineering Journal 359, 32-36

Engineering 11, 80-86

Chemical Engineering Journal 413, 127388

Chemical Engineering Journal 359, 32-36

Chemical Communications 54 (68), 9414-9417

Numerous other publications from the period spanning 2015 to 2019

The rationale behind the suitability of SOFOUR-DPDS-Ni for C₄ isomer separation instead of C₂H₆/CO₂ separation isn't expounded upon. Conversely, it might be beneficial to investigate whether SOFOUR-1-Zn and SOFOUR-TEPE-Zn were tested for their capability to separate C₄H₆/Iso-C₄H₈ and n-C₄H₈/iso-C₄H₈. Why then SOFOUR-DPDS-Ni is needed

This submission seems to imply that SOFOUR-DPDS-Ni, a new compound, exhibits a certain level of efficiency in separating C₄H₆/Iso-C₄H₈ and n-C₄H₈/iso-C₄H₈. However, the authors seem to have intentionally omitted a connection to previously identified sulfate-pillared ultramicroporous materials, giving the impression of an entirely novel discovery. Notably, the authors don't acknowledge the existence of the initial molecular sieving MOF for n-C₄H₁₀/iso-C₄H₁₀ and n-C₅H₁₂/iso-C₅H₁₂ (Angewandte Chemie International Edition 54 (48), 14353-14358, Chemical Society Reviews 46 (11), 3402-3430). In this work, the discussion primarily revolves around zeolites (specifically DDR), with the claim that no MOF molecular sieves were reported prior.

Given the aforementioned points, it is arguable that this work may not align well with the standards expected for publication in Nature Communications.

Reviewer #3 (Remarks to the Author):

In their study „Molecular sieving of iso-butene from C4 olefins with simultaneous high 1,3-butadiene and n-butene uptakes on a novel sulfate-pillared adsorbent” Liu and coworkers describe the synthesis of a sulfate pillared adsorbent from NiSO₄ and 4,4'-dipyridyldisulfide (DPDS) in methanol. The framework features an interesting structure, similar to the known class of coordination polymers with an interdigitated structure (CIDs). Ni²⁺ are bridged by two DPDS molecules, forming a 1D Ni(DPDS)₂ chain. These chains are then pillared in the apical position of the Ni by the sulfate anions, forming 2D sheets of the composition Ni(SO₄)(DPDS)₂. The crystal structure was solved through Rietveld Refinement of powder X-ray diffraction patterns. The gas was then analyzed towards its gas adsorption properties including CO₂ and N₂ Adsorption isotherms for determination of porosity. Furthermore, an extensive analysis of the adsorption and separation properties towards butadiene, n-butene and iso-butene is done. The authors performed single component isotherms, evaluated the heats of adsorption and did very detailed breakthrough studies. Overall the materials show good (and kinetically fast) uptake for n-butene and butadiene, while isobutene is barely adsorbed.

The authors present overall a great and very thorough work, well performed, with a lot of data – the concept of inhibiting the layer sliding by using a bend pillar seems quite innovative, inhibiting the layers to slide into each other in a interdigitated fashion. This work can be accepted for publication in Nature Communications after revisions and taking the comments listed underneath into account.

Comments:

According to the authors the material has a pore size of 4.7 Å, however the material does not show any uptake of N₂ which has a smaller kinetic diameter than the pore size. Can the authors give a reasoning for this?

The authors should give the sum formula of the compound in the text of the, which presumably is Ni(SO₄)(DPDS)₂.

The simulated powder pattern and the measured powder patterns in Figure S3 have quite some large differences, can the authors comment on that? It does not really reflect the high phase purity the authors are claiming in the manuscript. Also in the Rietveld Fit two low angle reflections are not included in the fit.

The authors state that this is a novel MOF. However many of the standard characterization data for novel, unreported compounds are missing. The authors should provide elemental analysis, IR Spectra and some SEM micrographs to asses the crystal size etc.

The authors are highlighting the use of a bend pillar, which enhances the selectivity. (i.e. since the frameworks with a linear pillar would interdigitate and show layer sliding according to the authors) – do the authors have any proof for this claim? I.e. can they reference something or show experiments on a Ni(SF₆)(DPDS)₂ or Ni(BF₄)(DPDS)₂?

Why is the disulfide in the cis and not the trans-conformation in the structure?

Response to Reviewers' Comments

Title: Molecular sieving of iso-butene from C₄ olefins with simultaneous high 1,3-butadiene and *n*-butene uptakes on a novel sulfate-pillared adsorbent

Manuscript ID: NCOMMS-23-21730

Corresponding Author: Prof. Jun Wang

Reviewer 1: *In this work, Wang and co-workers reported a novel sulfate-pillared adsorbent, SOFOUR-DPDS-Ni (DPDS=4,4'-dipyridyldisulfide), for efficient separation of iso-C₄H₈ from both binary and ternary C₄ olefin mixtures. The suitable pore sizes and shapes facilitate the selective adsorption of C₄H₆ and *n*-C₄H₈ over iso-C₄H₈, achieving remarkable Henry's selectivity for C₄H₆/iso-C₄H₈ and *n*-C₄H₈/iso-C₄H₈. Breakthrough experiments confirmed the feasibility of producing high-purity iso-C₄H₈ directly, and obtaining high-purity C₄H₆ and *n*-C₄H₈ during desorption processes. This manuscript is well-organized, and the structure of adsorbent is interesting with excellent performance that provide some useful clues for designing high performance porous materials for challenging gas separation. I suggest it to be published in **Nature Communications** after the following revisions.*

Author Response: We thank Reviewer #1 for the positive and valuable comments.

Comment 1: *More characterization of SOFOUR-DPDS-Ni, such as EA, FT-IR, and SEM, are suggested.*

Author Response: Thank you for the valuable comment. We have added the recommended characterizations such as EA, FT-IR, and SEM. The corresponding discussions are added in the revised *Manuscript* and *Supplementary Information*.

Modifications:

Manuscript: Page 5

The ultimate elemental analysis demonstrated that the element composition of each element corresponded well with the theoretical formula of SOFOUR-DPDS-Ni (C₂₀H₁₆N₄O₄S₅Ni, Supplementary Table 1). For instance, the measured content ratio of

N/S (0.39) was closely matched to the theoretical value (0.35). The scanning electron microscopy (SEM) image revealed a block morphology of SOFOUR-DPDS-Ni (Supplementary Fig. 12). Fourier transform infrared spectroscopy (FT-IR) spectra exhibited characteristic peaks corresponding to stretching vibrations for Ni-O at 493.2cm^{-1} and S-O in SO_4^{2-} at 1058.3cm^{-1} . Detailed discussions regarding FT-IR results are presented below Supplementary Fig. 13.

Supplementary Information: Page 7

Supplementary Figure 12. SEM image. SEM image of SOFOUR-DPDS-Ni.

Supplementary Information: Page 8

Supplementary Figure 13. FT-IR spectrum. The FT-IR spectrum of SOFOUR-DPDS-Ni.

The broad stretching vibration of the -OH group at 3372.0 cm⁻¹ was attributed to the presence of CH₃OH molecules in the framework. The strong peak at 1588.1 cm⁻¹ corresponded to the stretching vibration of C=C in the pyridine ring. The characteristic peaks at 3093.3, 1482.5, and 1415.5 cm⁻¹ represented the bending vibration of C-H bonds in either CH₃OH molecules or pyridine rings. The weak peak at 1323.0 cm⁻¹ indicated the stretching vibration of C-O in CH₃OH molecules. The weak peak located at 1223.3 cm⁻¹ belonged to the stretching vibration of C-N in the pyridine ring. The peak at 1058.3 cm⁻¹ represented the antisymmetric stretching vibration of S-O in the SO₄²⁻ group. The peaks at 712.1 and 595.9 cm⁻¹ were assigned to the stretching vibrations of S-C and S-S, respectively. The peak at 493.2 cm⁻¹ corresponded to the Ni-O or Ni-N stretching vibration.

Supplementary Information: Page 22

Supplementary Table 1. Ultimate element analysis of SOFOUR-DPDS-Ni.

Elemental		C	H	N	S	N/S
mass	Measured	36.82	3.10	8.47	21.56	0.39
(wt. %)	Theoretical	40.34	2.69	9.41	26.93	0.35

Comment 2: *The kinetic curve of C₄H₆ in Fig. S12 is not well-fitted, which may affect the accuracy of the results.*

Author Response: Thanks for the comment. We retested the kinetic curve of C₄H₆ and re-fitted it with high accuracy.

Modifications:

Supplementary Information: Page 9

Supplementary Figure 14. Fitting of kinetic curve. Kinetic profile of (a) C₄H₆ and (b) n-C₄H₈ on SOFOUR-DPDS-Ni.

Comment 3: *The crystallinity and integrity of the SOFOUR-DPDS-Ni should be checked after adsorption tests and cycling breakthrough experiments.*

Author Response: Thanks for the comment. We have collected the PXRD patterns of SOFOUR-DPDS-Ni after cycled adsorption isotherms and breakthrough experiments, which have been added in the revised *Supplementary Information*. As shown in Supplementary Figure 37, the PXRD patterns of SOFOUR-DPDS-Ni did not change after various experiments, indicating its good structural stability.

Modifications:

Manuscript: Page 11

Furthermore, the PXRD patterns of SOFOUR-DPDS-Ni remained change after cycled adsorption tests and breakthrough experiments, thereby indicating its exceptional structural stability (Supplementary Fig. 39).

Supplementary Information: Page 21

Supplementary Figure 39. Crystallinity and integrity. PXRD patterns of SOFOUR-DPDS-Ni after cycled adsorption tests and breakthrough experiments.

Comment 4: Detailed information for calculating the productivity of *iso*-C₄H₈ and the dynamic adsorption capacity of C₄H₆ and *n*-C₄H₈ should be provided.

Author Response: Thanks for the comment. As suggested, the detailed information for calculating the productivity of *iso*-C₄H₈ and the dynamic adsorption capacity of C₄H₆ and *n*-C₄H₈ have been added to the *supplementary information*.

Modifications:

Supplementary Information: Page 20

The kinetic adsorption capacity (Q) of C₄H₆ or *n*-C₄H₈ is calculated as:

$$Q = \frac{v \times V\%}{m \times 22.4} \times \int_{t_1}^{t_2} (c_1 - c_i) dt = \frac{v \times V\%}{m \times 22.4} \times S_1$$

v is the flow rate of the gas mixture, $V\%$ is the molar fraction of C₄H₆ or *n*-C₄H₈, and m is the mass of the adsorbent.

Supplementary Figure 37. Schematic diagram of calculation. The calculation diagram of kinetic adsorption capacity of C₄H₆ or *n*-C₄H₈.

The productivity (q) of $iso-C_4H_8$ is calculated as:

$$q = \frac{v \times V\%}{m \times 22.4} \times \int_{t_3}^{t_4} (c_x - c_i) dt = \frac{v \times V\%}{m \times 22.4} \times S_2$$

v is the flow rate of the gas mixture, $V\%$ is the molar fraction of $iso-C_4H_8$, and m is the mass of the adsorbent.

Supplementary Figure 38. Schematic diagram of calculation. The calculation diagram of $iso-C_4H_8$ productivity.

Comment 5: The structure of $iso-C_4H_8$ molecule in Fig. 3i must be corrected.

Author Response: Thank you for pointing out this mistake. We have corrected the image of optimized $iso-C_4H_8$ in Figure 3i.

Modifications:

Manuscript: Page 10

Fig. 3 GCMC simulated and DFT-D calculated results in SOFOUR-DPDS-Ni.

Comment 6: The value of R_p and R_{wp} in Figure S2 are missing.

Author Response: Thanks for the advice. We have added the values of R_p and R_{wp} to Supplementary Figure 2 in the revised *Supplementary Information*.

Modifications:

Supplementary Information: Page 2

Supplementary Figure 1. Rietveld refinement. PXRD Rietveld refinement for SOFOUR-DPDS-Ni.

Comment 7: Fig. 2d and Fig. S9 are duplicates, please check.

Author Response: Thanks for the comment. We have removed the duplicate image in the original Figure S9.

Comment 8: The article should be formatted by the requirements of *Nat. Commun.*, such as references and methods.

Author Response: Thanks for the comment. We have revised our manuscript according to the format of *Nat. Commun.*

Comment 9: The description of comparing the pore size of SOFOUR-DPDS-Ni with C_4 olefins in Line 121 and 122 is wrong.

Author Response: Thanks for pointing out this mistake. We have corrected this error in the revised *Manuscript*.

Modifications:

Manuscript: Page 5

The pore sizes of SOFOUR-DPDS-Ni were larger than the kinetic diameters of C₄H₆ (4.31 Å) and *n*-C₄H₈ (4.46 Å), yet smaller than that of *iso*-C₄H₈ (4.84 Å), suggesting the potential molecular sieving of *iso*-C₄H₈ from C₄H₆ and *n*-C₄H₈ counterparts.

Comment 10: *Please pay attention to the layout of Tables in the Supplementary Information.*

Author Response: Thanks for the valuable comment. We have rearranged the Tables in the revised *Supplementary Information*.

Referee 2: *This paper presents an innovative reticular chemistry modification applied to SOFOUR-1-Zn and SOFOUR-TEPE-Zn, which were originally investigated for C₂H₂/CO₂ separation. This modification results in the creation of a novel material termed SOFOUR-DPDS-Ni, now studied for C₄H₆/*iso*-C₄H₈ and *n*-C₄H₈/*iso*-C₄H₈ separation. The discovery of SOFOUR-DPDS-Ni is undoubtedly noteworthy. However, it's worth noting that the transition to higher-density sulfate-pillared materials, or materials with equal density, for testing C₄ isomers isn't entirely novel, as it resonates with prior publications within this field. Examples of these references include:*

Chemical Engineering Journal 359, 32-36

Engineering 11, 80-86

Chemical Engineering Journal 413, 127388

Chemical Engineering Journal 359, 32-36

Chemical Communications 54 (68), 9414-9417

Numerous other publications from the period spanning 2015 to 2019

The rationale behind the suitability of SOFOUR-DPDS-Ni for C₄ isomer separation instead of C₂H₂/CO₂ separation isn't expounded upon. Conversely, it might be

beneficial to investigate whether SOFOUR-1-Zn and SOFOUR-TEPE-Zn were tested for their capability to separate C₄H₆/iso-C₄H₈ and n-C₄H₈/iso-C₄H₈. Why then SOFOUR-DPDS-Ni is needed.

This submission seems to imply that SOFOUR-DPDS-Ni, a new compound, exhibits a certain level of efficiency in separating C₄H₆/iso-C₄H₈ and n-C₄H₈/iso-C₄H₈. However, the authors seem to have intentionally omitted a connection to previously identified sulfate-pillared ultra-microporous materials, giving the impression of an entirely novel discovery. Notably, the authors don't acknowledge the existence of the initial molecular sieving MOF for n-C₄H₁₀/iso-C₄H₁₀ and n-C₅H₁₂/iso-C₅H₁₂ (Angewandte Chemie International Edition 54 (48), 14353-14358, Chemical Society Reviews 46 (11), 3402-3430). In this work, the discussion primarily revolves around zeolites (specifically DD3R), with the claim that no MOF molecular sieves were reported prior.

Given the aforementioned points, it is arguable that this work may not align well with the standards expected for publication in Nature Communications.

Author Response: Thank you for the valuable comments, which will significantly improve the quality of our work. We have realized that the issues mentioned were not fully or properly addressed in the original manuscript. Therefore, this work has been revised accordingly.

We totally agree that SOFOUR-1-Zn and SOFOUR-TEPE-Zn are two presentive SO₄²⁻-pillared MOFs for efficient C₂H₂/CO₂ separation. Gladly, SOFOUR-TEPE-Zn was reported by our group in *Adv. Mater.* These two MOFs have been discussed and cited in the revised *Manuscript* (Page 3, highlighted in Green). In comparison, SOFOUR-1-Zn and SOFOUR-TEPE-Zn were synthesized using four-connected ligands (2,4,5-tetra (4-pyridyl) benzene and 1,1,2,2-tetra(pyridin-4-yl) ethene) and octahedron SiF₆²⁻ anions, affording *fsc* topology. Whereas, SOFOUR-DPDS-Ni was prepared using a V-shape two-connected ligand (DPDS) and tetrahedral SO₄²⁻ pillars resulting in *sql* topology. Furthermore, previously reported MOFs with *sql* topology commonly exhibited structural flexibility as observed in UTSA-300 (*J. Am. Chem. Soc.*, 2017, 139, 8022), NCU-100 (*Nat. Commun.*, 2022, 13, 200), and ZUL-220 (*Nat.*

Commun., 2020, 11, 6259). However, the flexibility often leads to high gate-opening pressures, which can cause co-adsorption and limited diffusion rate, especially for relatively large C₄ molecules. In sharp contrast, the avoidance of sliding interlayers by SO₄²⁻ pillars enables the preservation of pore shapes and sizes during activation and adsorption processes, thereby achieving the molecular sieving of *iso*-C₄H₈ from C₄H₆ and *n*-C₄H₈ counterparts. Therefore, the design and rationale behind SOFOUR-DPDS-Ni represent novelty.

As recommended, we also conducted experiments to demonstrate the performance disparities between SOFOUR-1-Zn/SOFOUR-TEPE-Zn and SOFOUR-DPDS-Ni. Firstly, the adsorption isotherms of SOFOUR-DPDS-Ni exhibited a high C₂H₂ uptake (2.87 mmol g⁻¹) and moderate CO₂ uptake (1.36 mmol g⁻¹) at 298 K and 1 bar (Figure R1), which was significantly inferior compared to the molecular sieving on SOFOUR-TEPE-Zn. Secondly, we measured the adsorption isotherms of C₄ olefins on SOFOUR-1-Zn and SOFOUR-TEPE-Zn at 298 K (Supplementary Figure 17). Comparable uptakes for *n*-C₄H₈ (0.72 mmol g⁻¹) and *iso*-C₄H₈ (0.55 mmol g⁻¹) were observed on SOFOUR-1-Zn adsorbed. Meanwhile, negligible *n*-C₄H₈ and *iso*-C₄H₈ uptakes (< 0.13 mmol g⁻¹) were detected on SOFOUR-TEPE-Zn. These inferior performances on SO₄²⁻-pillared adsorbents further emphasize the advantages of pore environments of SOFOUR-DPDS-Ni. Detailed discussions have been added in the revised *Manuscript*.

We acknowledge that the previous literature on the separations of C₄ olefin isomers. We appreciate the recommended references and have thoroughly studied these studies. Prof. Eddaoudi and Prof. Xing *et al.* reported MFFIVE-1-Ni adsorbents for C₂H₂/C₂H₄ and CO₂/C₂H₂ separations (*Chem. Eng. J.*, 2019, 359, 32-36). For the first time, Prof. Xing's group designed ZU-36-Ni and ZU-36-Fe to separate *trans*-/*cis*-2-butene mixture (*Engineering.*, 2022, 11, 80-86). Prof. Eddaoudi's group synthesized the first zeolite-like metal-organic framework (ZMOF) with *ana* topology and a new rare-earth metal (RE) fcu-MOF with a suitable aperture size (4.7 Å) for *n*-C₄H₁₀/*iso*-C₄H₁₀ and *n*-C₅H₁₂/*iso*-C₅H₁₂ separation (*Chem. Commun.*, 2018, 54, 9414-9417 and *Angew. Chem. Int. Ed.*, 2015, 54, 14353-14358). Prof. Belmabkhout *et al.* reported Y-fum-**fcu**-MOF

containing octahedral and tetrahedral cages with triangular pore aperture size (≈ 4.7 Å) for C₄ olefins separations (*Chem. Eng. J.*, 2021, 413, 127388). Obviously, these important works provided guidance for our material design. We have cited these references as references 5-7 and 21-23 in the revised *Manuscript* and reference 3 in *Supplementary Information*. After conducting a comprehensive background review, we found that only a few adsorbents could achieve molecular sieving of *iso*-C₄H₈ (TMOF-1, *Chem. Eng. J.*, 2021, 425, 130580 and ZU-52, *Angew. Chem. Int. Ed.*, 2017, 56, 16282-16287). Therefore, SOFOUR-DPDS-Ni exhibited efficient *iso*-C₄H₈ molecular-sieving separation with simultaneous high C₄H₆ and *n*-C₄H₈ uptakes at low pressure and 1.0 bar, which is meaningful and presents progress.

Besides the conventional DD3R zeolite, we agree that only a few molecular sieving MOFs for isomer paraffins (*e.g.*, *n*-C₄H₁₀/*iso*-C₄H₁₀ and *n*-C₅H₁₂/*iso*-C₅H₁₂) have been previously reported. We would like to thank you for providing the valuable reference published by Prof. Eddaoudi's group (*Angew. Chem. Int. Ed.*, 2015, 54, 14353), Y-fum demonstrated excellent molecular sieving capabilities for *iso*-C₄H₁₀ and *iso*-C₅H₁₂ over their corresponding *n*-paraffins. This important work has been discussed in the revised *Manuscript*. We hope the modifications adequately address previous works on SO₄²⁻-pillared adsorbents and C₄ separation performances.

Modifications:

Manuscript: Page 2 and Page 3

Physisorption utilizing porous adsorbents, *e.g.*, zeolites and metal-organic frameworks (MOFs)⁵⁻¹³, shows great promise in various challenging gas separations, including C₂H₂/C₂H₄, CO₂/C₂H₂, and *n*-C₄H₁₀/*iso*-C₄H₁₀¹⁴⁻²².

To date, achieving complete molecular sieving of *iso*-C₄H₈ remains challenging for MOF adsorbents^{1,25}. For example, Prof. Eddaoudi's group reported Y-fum with a suitable aperture size of ~ 4.7 Å exhibited excellent molecular sieving capabilities for *iso*-C₄H₁₀ and *iso*-C₅H₁₂ from their corresponding *n*-paraffins²³.

Manuscript: Page 5

Additionally, the adsorption isotherms exhibited a high C₂H₂ uptake (2.87 mmol g⁻¹)

and the moderate CO₂ uptake (1.36 mmol g⁻¹) at 298 K and 1.0 bar (Supplementary Fig. 9). The C₂H₂/CO₂ separation performance was considerably inferior compared to the molecular sieving effect on SO₄²⁻-pillared SOFOUR-TEPE-Zn³¹.

Manuscript: Page 7

In comparison, we measured the adsorption isotherms of C₄ olefins on SOFOUR-1-Zn and SOFOUR-TEPE-Zn at 298 K (Supplementary Fig. 17). SOFOUR-1-Zn adsorbed comparable uptakes for *n*-C₄H₈ (0.72 mmol g⁻¹) and *iso*-C₄H₈ (0.55 mmol g⁻¹). Meanwhile, negligible *n*-C₄H₈ and *iso*-C₄H₈ uptakes (< 0.13 mmol g⁻¹) were observed on SOFOUR-TEPE-Zn. These inferior performances on SO₄²⁻-pillared adsorbents further highlighted the advantages of pore environments of SOFOUR-DPDS-Ni.

Manuscript: Page 19 and 21

5. Adil, K. *et al.* Gas/vapour separation using ultra-microporous metal–organic frameworks: insights into the structure/separation relationship. *Chem. Soc. Rev.* **46**, 3402–3430 (2017).

6. Belmabkhout, Y. *et al.* Hydrocarbon recovery using ultra-microporous fluorinated MOF platform with and without uncoordinated metal sites: I- structure properties relationships for C₂H₂/C₂H₄ and CO₂/C₂H₂ separation. *Chem. Eng. J.* **359**, 32–36 (2019).

7. Mohideen, M. I. H. *et al.* Upgrading gasoline to high octane numbers using a zeolite-like metal–organic framework molecular sieve with **ana**-topology. *Chem. Commun.* **54**, 9414–9417 (2018).

21. Assen, A. H. *et al.* Kinetic separation of C₄ olefins using Y-fum-fcu-MOF with ultra-fine-tuned aperture size. *Chem. Eng. J.* **413**, 127388 (2021).

22. Zhang, Z. *et al.* Efficient splitting of trans-/cis-olefins using an anion-pillared ultramicroporous metal–organic framework with guest-adaptive pore channels. *Engineering* **11**, 80–86 (2022).

23. Assen, A. H. *et al.* Ultra-tuning of the rare-earth fcu-MOF aperture size for selective molecular exclusion of branched paraffins. *Angew. Chem. Int. Ed.* **54**, 14353–14358 (2015).

Supplementary Information: Page 24

Supplementary Table 6. Comparison of separation selectivity based on uptake ratio of

reported materials.

Materials	Temperature (K)	Pressure (kPa)	Uptake selectivity			Reference
			C ₄ H ₆ / n -C ₄ H ₈	C ₄ H ₆ / iso -C ₄ H ₈	n -C ₄ H ₈ / iso -C ₄ H ₈	
SOFOUR-DPDS-Ni	298	100	1.14	10.50	9.25	This work
TMOF-1	298	100	1.5	8.2	5.4	1
ZU-619	298	100	2.51	4.65	1.85	1
Mn-bpdc	298	101	40	45	1.1	2
Y-fum-fcu-MOF	303	101	0.97	2.66	2.76	3
Mg-gallate	298	101	1.3	15.1	11.2	4
Co-gallate	298	101	2.1	14.3	6.8	4
Ni-gallate	298	101	2.4	15.4	6.3	4

.....

Supplementary References

3. Assen, A. H. *et al.* Kinetic separation of C₄ olefins using Y-fum-fcu-MOF with ultra-fine-tuned aperture size. *Chem. Eng. J.* **413**, 127388 (2021).

Figure R1. The adsorption isotherms of C₂H₂ and CO₂ by SOFOUR-DPDS-Ni.

Supplementary Figure 17. Adsorption isotherms. Adsorption isotherms of C₄ olefins on (a) SOFOUR-1-Zn and (b) SOFOUR-TEPE-Zn at 298 K.

Reviewer 3: *In their study, “Molecular sieving of iso-butene from C₄ olefins with simultaneous high 1,3-butadiene and n-butene uptakes on a novel sulfate-pillared adsorbent”, Liu and coworkers describe the synthesis of a sulfate pillared adsorbent from NiSO₄ and 4,4'-dipyridyldisulfide (DPDS) in methanol. The framework features an interesting structure, similar to the known class of coordination polymers with an interdigitated structure (CIDs). Ni²⁺ are bridged by two DPDS molecules, forming a 1D Ni(DPDS)₂ chain. These chains are then pillared in the apical position of the Ni by the sulfate anions, forming 2D sheets of the composition Ni(SO₄)(DPDS)₂. The crystal structure was solved through Rietveld Refinement of powder X-ray diffraction patterns. The gas was then analyzed towards its gas adsorption properties including CO₂ and N₂ Adsorption isotherms for determination of porosity. Furthermore, an extensive analysis of the adsorption and separation properties towards butadiene, n-butene and iso-butene is done. The authors performed single component isotherms, evaluated the heats of adsorption and did very detailed breakthrough studies. Overall, the materials show good (and kinetically fast) uptake for n-butene and butadiene, while isobutene is barely adsorbed.*

The authors present overall a great and very thorough work, well performed, with a lot of data – the concept of inhibiting the layer sliding by using a bend pillar seems quite innovative, inhibiting the layers to slide into each other in an interdigitated fashion.

This work can be accepted for publication in Nature Communications after revisions and taking the comments listed underneath into account.

Author Response: We sincerely thank Reviewer 3's positive assessment. We have carefully addressed the following comments and revised our manuscript accordingly.

Comment 1: *According to the authors the material has a pore size of 4.7 Å, however the material does not show any uptake of N₂, which has a smaller kinetic diameter than the pore size. Can the authors give a reasoning for this?*

Author Response: Thank you for the valuable comment. We determined the pore size of SOFOUR-DPDS-Ni to be 4.7 Å through the adsorption isotherm of CO₂ (3.3 Å) at 195 K. Whereas, the slightly larger N₂ (3.65 Å) cannot enter the pore channels of SOFOUR-DPDS-Ni. These results are consistent with the structure-derived pore sizes (3.3×4.5 and 3.6×3.7 Å²) obtained using a probe size of 0.1 Å on the accessible Connolly surface (Supplementary Fig. 5). Furthermore, at the low temperature of 77 K, benzene rings exhibit limited rotational motion.

Comment 2: *The authors should give the sum formula of the compound in the text of the, which presumably is Ni(SO₄)(DPDS)₂.*

Author Response: Thanks for the advice. We have provided the chemical formula of Ni(DPDS)₂SO₄ for SOFOUR-DPDS-Ni in the revised *Manuscript*.

Modifications:

Manuscript: Page 4

The reaction of NiSO₄·6H₂O and DPDS in methanol solutions at room temperature yielded light blue powder of SOFOUR-DPDS-Ni with the chemical formula of Ni(DPDS)₂SO₄ (Fig. 1c, see methods for details).

Manuscript: Page 13

The material was designated as SOFOUR-DPDS-Ni, with a chemical formula of Ni(DPDS)₂SO₄.

Comment 3: *The simulated powder pattern and the measured powder patterns in Figure S3 have quite some large differences, can the authors comment on that? It does not really reflect the high phase purity the authors are claiming in the manuscript. Also, in the Rietveld Fit two low angle reflections are not included in the fit.*

Author Response: Thank you for the valuable comment. We did realize that the small peaks at low XRD angles are missing compared to the simulated pattern. This phenomenon can be attributed to the presence of microscopic imperfections and the accumulation of grains in powder samples, resulting in the extinction of certain crystal planes. This phenomenon is not occasional in micro-crystal MOFs; for example, Ni(4-DPDS)₂MoO₄ in Figure R2 and ZNU-9 in Figure R3 also missed some peaks at low angles (*Angew. Chem. Int. Ed.*, 2022, e202116686; *Angew. Chem. Int. Ed.*, 2023, e202309925). Similar situations have been reported in other literatures (*Chem. Commun.*, 2015, 51, 5610; *J. Mater. Chem. A.*, 2023, 11, 17821; *Angew. Chem. Int. Ed.*, 2022, 61, e202200947). In this work, the R_p and R_{wp} values obtained from PXRD Rietveld refinement are reasonable and consistent with the simulated pattern for the main PXRD peaks of our synthesized sample, indicating the accuracy of our predicted structure.

Figure R2. PXRD patterns of Ni(4-DPDS)₂MoO₄.

Figure R3. PXRD patterns of ZNU-9.

Comment 4: *The authors state that this is a novel MOF. However, many of the standard characterization data for novel, unreported compounds are missing. The authors should provide elemental analysis, IR Spectral and some SEM micrographs to assess the crystal size etc.*

Author Response: Thank you for the valuable comment. We have added the recommended characterizations such as EA, FT-IR, and SEM. The corresponding discussions are added in the revised *Manuscript* and *Supplementary Information*.

Modifications:

Manuscript: Page 5

The ultimate elemental analysis demonstrated that the element composition of each element corresponded well with the theoretical formula of SOFOUR-DPDS-Ni ($C_{20}H_{16}N_4O_4S_5Ni$, Supplementary Table 1). For instance, the measured content ratio of N/S (0.39) was closely matched to the theoretical value (0.35). The scanning electron microscopy (SEM) image revealed a block morphology of SOFOUR-DPDS-Ni (Supplementary Fig. 12). Fourier transform infrared spectroscopy (FT-IR) spectra exhibited characteristic peaks corresponding to stretching vibrations for Ni-O at

493.2 cm^{-1} and S-O in SO_4^{2-} at 1058.3 cm^{-1} . Detailed discussions regarding FT-IR results are presented below Supplementary Fig. 13.

Supplementary Information: Page 7

Supplementary Figure 12. SEM image. SEM image of SOFOUR-DPDS-Ni.

Supplementary Information: Page 8

Supplementary Figure 13. FT-IR spectrum. The FT-IR spectrum of SOFOUR-DPDS-Ni.

The broad stretching vibration of the -OH group at 3372.0 cm^{-1} was attributed to the presence of CH_3OH molecules in the framework. The strong peak at 1588.1 cm^{-1} corresponded to the stretching vibration of $\text{C}=\text{C}$ in the pyridine ring. The characteristic peaks at 3093.3, 1482.5, and 1415.5 cm^{-1} represented the bending vibration of C-H

bonds in either CH₃OH molecules or pyridine rings. The weak peak at 1323.0 cm⁻¹ indicated the stretching vibration of C-O in CH₃OH molecules. The weak peak located at 1223.3 cm⁻¹ belonged to the stretching vibration of C-N in the pyridine ring. The peak at 1058.3 cm⁻¹ represented the antisymmetric stretching vibration of S-O in the SO₄²⁻ group. The peaks at 712.1 and 595.9 cm⁻¹ were assigned to the stretching vibrations of S-C and S-S, respectively. The peak at 493.2 cm⁻¹ corresponded to the Ni-O or Ni-N stretching vibration.

Supplementary Information: Page 22

Supplementary Table 1. Ultimate element analysis of SOFOUR-DPDS-Ni.

Elemental		C	H	N	S	N/S
mass	Measured	36.82	3.10	8.47	21.56	0.39
(wt. %)	Theoretical	40.34	2.69	9.41	26.93	0.35

Comment 5: *The authors are highlighting the use of a bend pillar, which enhances the selectivity. (i.e. since the frameworks with a linear pillar would interdigitate and show layer sliding according to the authors) – do the authors have any proof for this claim? I.e. can they reference something or show experiments on a Ni(SiF₆)(DPDS)₂ or Ni(BF₄)(DPDS)₂?*

Author Response: Thanks for the valuable comment. In recent years, MOF adsorbents featuring linear pillars (e.g., SiF₆²⁻ and GeF₆²⁻), such as UTSA-300 (Figure R4, *J. Am. Chem. Soc.*, 2017, 139, 8022–8028) and NCU-100 (Figure R6, *Nat. Commun.*, 2022, 13, 200), have demonstrated layer-to-layer sliding after removing solvents. Their C₂H₂ adsorption isotherms exhibited a pronounced “gate-opening” effect due to the rotational motion of linear pillars and ligand flipping (Figure R5 and Figure R7). To validate the rigidity in angular SO₄²⁻ pillared SOFOUR-DPDS-Ni, we first recorded the C₂H₂, C₂H₄, and CO₂ adsorption isotherms. Figure R8 shows no discernable “gate-opening” effect, indicating the absence of structure flexibility.

Second, as recommended, we attempted to synthesize Ni(SiF₆)(DPDS)₂ and Ni(BF₄)₂(DPDS)₂. Unfortunately, the PXRD patterns of these two materials did not

match the simulated structure (Figure R9a and b), indicating that linear pillars cannot prepare MOFs with DPDS ligands and Ni ions. The adsorption isotherms for C_4H_6 , n - C_4H_8 , and *iso*- C_4H_8 also displayed negligible uptakes (Figure R9c and d). Notably, we have successfully synthesized a novel structure of $Cu(BF_4)_2(DPDS)_2$ (Figure R10, CCDC: 2162794). The C_2H_4 and C_3H_6 adsorption isotherms exhibited the characteristic “gate-opening” effect, demonstrating its structural flexibility (Figure R11). This work is currently under revision, that I am unable to disclose all the research data at this time.

Figure R4. The structure of UTSA-300 before (a) and after activation (b).

Figure R5. C_2H_2 , CO_2 , and C_2H_4 sorption isotherms for UTSA-300a.

Figure R6. The structure of NCU-100 before (a) and after activation (b).

Figure R7. C_2H_2 and CO_2 sorption isotherms for NCU-100.

Figure R8. Adsorption isotherms of SOFOUR-DPDS-Ni for various gases.

Figure R9. Structure of simulated 2D Ni(XF₆)(DPDS)₂ (a); The PXRD patterns of as-synthesized Ni(SiF₆)(DPDS)₂ and Ni(BF₄)₂(DPDS)₂ (b); The adsorption isotherms of C₄ olefins on Ni(SiF₆)(DPDS)₂ and Ni(BF₄)₂(DPDS)₂ (d) at 298 K.

Figure R10. The PXRD patterns of Cu(BF₄)₂(DPDS)₂.

Figure R11. The adsorption isotherms of C₂H₄ (a) and C₃H₆ (b) on Cu(BF₄)₂(DPDS)₂ at 298 K.

Comment 6: *Why is the disulfide in the cis and not the trans-conformation in the structure?*

Author Response: Thank you for the valuable comment. During the experiment, we also noticed potential conformational changes in *cis* and *trans* forms of DPDS (Figure R12). Theoretically, a three-dimensional pcu framework should be obtained using *trans* DPDS ligand. However, after numerous attempts (such as changing synthesis temperature, times, and solvent, using mixed solvent *etc.*), we were unable to obtain any three-dimensional structural material. Meanwhile, the energy barrier for the rotation of the -S-S- bond is moderate (*Org. Lett.*, 2001, 3, 3639-3641), and it is prone to form a twisted *cis* configuration (*Inorg. Chem.*, 2001, 40, 2430-2433). We also calculated the energy level of DPDS ligand by Dmol3, which was -1291.05 and -1291.03 Ha for *cis* and *trans* conformations, respectively. The lower energy of the twisted *cis* configuration indicates its more stable structure. Moreover, the reported MOFs with DPDS ligand were also found to exhibit a *cis* configuration (*Angew. Chem. Int. Ed.*, 2022, e202116686).

Figure R12. Twisted *cis* (a) and *trans* (b) configuration of DPDS.

REVIEWERS' COMMENTS

Reviewer #1 (Remarks to the Author):

The quality of this manuscript has been greatly improved therefore this work is suitable for publication.

Reviewer #2 (Remarks to the Author):

The Authors have clearly been responsive in the process of revising their manuscript.

I still have one observation that will be good to clarify before the acceptance of this manuscript. Looking at Fig 4C, it looks like that C₄H₆ separation from iso-C₄H₈ (this last is actually barely adsorbed) is affected by the presence of n-C₄H₈ (1-Butene). Any mechanistic explanation about this loss in performance ?

The authors have also thoroughly addressed the concerns raised by reviewer 3.

Reviewer #1 (Remarks to the Author):

The quality of this manuscript has been greatly improved therefore this work is suitable for publication.

Author Response: We thank reviewer #1 for the valuable comments and positive recommendation.

Reviewer #2 (Remarks to the Author):

Comment 1: The Authors have clearly been responsive in the process of revising their manuscript. I still have one observation that will be good to clarify before the acceptance of this manuscript. Looking at Fig 4C, it looks like that C₄H₆ separation from iso-C₄H₈ (this last is actually barely adsorbed) is affected by the presence of n-C₄H₈ (1-Butene). Any mechanistic explanation about this loss in performance?

Author Response: We appreciate the valuable comment. Base on the static adsorption isotherm (Fig 2a), it can be observed that iso-C₄H₈ and C₄H₆ exhibit similar adsorption behaviors when interacting with SOFOUR-DPDS-Ni. Therefore, in the ternary gas-mixture breakthrough experiment, their competitive adsorption will inevitably influence the dynamic adsorption uptakes and separation performances.

Modifications:

Manuscript: Page 11

Fig. 4c depicted that iso-C₄H₈ was immediately detected, while C₄H₆ and n-C₄H₈ were retained in the column for 46.8 min and 43.2 min, respectively. Their competitive adsorptions will simultaneously occupy the available adsorption sites and inevitably influence their adsorption uptakes.

Comment 2: The authors have also thoroughly adressed the concerns raised by reviewer 3.

Author Response: We thank reviewer #1 for the positive recommendation.